# Alleviating tiling effect by random walk sliding window in high-resolution histological whole slide image synthesis

**Shunxing Bao**[1]                                                                              SHUNXING.BAO@VANDERBILT.EDU
**Ho Hin Lee**[2]                                                                                 HO.HIN.LEE@VANDERBILT.EDU
**Qi Yang**[2]                                                                                    QI.YANG@VANDERBILT.EDU
**Lucas W. Remedios**[2]                                                          LUCAS.W.REMEDIOS@VANDERBILT.EDU
**Ruining Deng**[2]                                                                             R.DENG@VANDERBILT.EDU
**Can Cui**[2]                                                                                 CAN.CUI.1@VANDERBILT.EDU
**Leon Y. Cai**[3]                                                                            LEON.Y.CAI@VANDERBILT.EDU
**Kaiwen Xu**[2]                                                                              KAIWEN.XU@VANDERBILT.EDU
**Xin Yu**[2]                                                                                    XIN.YU@VANDERBILT.EDU
**Sophie Chiron**[4]                                                                           SOPHIE.CHIRON@VUMC.ORG
**Yike Li**[5]                                                                                   YIKE.LI.1@VUMC.ORG
**Nathan Heath Patterson**[6]                                               NATHAN.H.PATTERSON@VANDERBILT.EDU
**Yaohong Wang**[7]                                                          YAOHONG.WANG.1@VANDERBILT.EDU
**Jia Li**[8]                                                                                    JIA.LI.1@VUMC.ORG
**Qi Liu**[8,9]                                                                                  QI.LIU@VUMC.ORG
**Ken S. Lau**[9,10,11]                                                                       KEN.S.LAU@VANDERBILT.EDU
**Joseph T. Roland**[12]                                                                  JOSEPH.T.ROLAND@VUMC.ORG
**Lori A. Coburn**[4,7,13,14]                                                              LORI.COBURN@VUMC.ORG
**Keith T. Wilson**[4,7,13,14,15]                                                          KEITH.WILSON@VUMC.ORG
**Bennett A. Landman**[1,2,3]                                                       BENNETT.LANDMAN@VANDERBILT.EDU
**Yuankai Huo**[1,2]                                                                       YUANKAI.HUO@VANDERBILT.EDU

[1] *Electrical and Computer Engineering, Vanderbilt University, Nashville, TN, USA*

[2] *Department of Computer Science, Vanderbilt University, Nashville, TN, USA*

[3] *Department of Biomedical Engineering, Vanderbilt University, Nashville, TN, USA*

[4] *Division of Gastroenterology, Hepatology, and Nutrition, Department of Medicine, Vanderbilt University Medical Center, Nashville, TN, USA*

[5] *Department of Otolaryngology-Head and Neck Surgery, Vanderbilt University Medical Center, Nashville, TN, USA*

[6]*Mass Spectrometry Research Center, Vanderbilt University, Nashville, TN, USA*

[7] *Department of Pathology, Microbiology and Immunology, Vanderbilt University Medical Center, Nashville, TN, USA*

[8] *Dept. of Biostatistics, Vanderbilt University Medical Center, Nashville, TN, USA*

[9] *Center for Quantitative Sciences, Vanderbilt University Medical Center, Nashville, TN, USA*

[10] *Epithelial Biology Center, Vanderbilt University Medical Center, Nashville, TN, USA*

[11] *Dept. of Cell and Developmental Biology, Vanderbilt University School of Medicine, Nashville, TN, USA*

[12] *Epithelial Biology Center, Vanderbilt University Medical Center, Nashville, TN, USA*

[13] *Vanderbilt Center for Mucosal Inflammation and Cancer, Nashville, TN, USA*

[14] *Veterans Affairs Tennessee Valley Healthcare System, Nashville, TN, USA*

[15] *Program in Cancer Biology, Vanderbilt University School of Medicine, Nashville, TN, USA*

**Editors:** Accepted for publication at MIDL 2023

## Abstract

Multiplex immunofluorescence (MxIF) is an advanced molecular imaging technique that can simultaneously provide biologists with multiple (i.e., more than 20) molecular markers on a single histological tissue section. Unfortunately, due to imaging restrictions, the more routinely used hematoxylin and eosin (H&E) stain is typically unavailable with MxIF on the same tissue section. As biological H&E staining is not feasible, previous efforts have been made to obtain H&E whole slide image (WSI) from MxIF via deep learning empowered virtual staining. However, the tiling effect is a long-lasting problem in high-resolution WSI-wise synthesis. The MxIF to H&E synthesis is no exception. Limited by computational resources, the cross-stain image synthesis is typically performed at the patch-level. Thus, discontinuous intensities might be visually identified along with the patch boundaries assembling all individual patches back to a WSI. In this work, we propose a deep learning based unpaired high-resolution image synthesis method to obtain virtual H&E WSIs from MxIF WSIs (each with 27 markers/stains) with reduced tiling effects. Briefly, we first extend the CycleGAN framework by adding simultaneous nuclei and mucin segmentation supervision as spatial constraints. Then, we introduce a random walk sliding window shifting strategy during the optimized inference stage, to alleviate the tiling effects. The validation results show that our spatially constrained synthesis method achieves a 56% performance gain for the downstream cell segmentation task. The proposed inference method reduces the tiling effects by using 50% fewer computation resources without compromising performance. The proposed random sliding window inference method is a plug-and-play module, which can be generalized for other high-resolution WSI image synthesis applications. The source code with our proposed model are available at
https://github.com/MASILab/RandomWalkSlidingWindow.git

**Keywords:** Unpaired synthesis, tiling effect, spatial constraints

## 1. Introduction

Multiplex immunofluorescence (MxIF) is a recently developed imaging platform that allows multiple (i.e., more than 20) cellular and histological markers to be investigated on a single tissue section (Vahadane et al., 2023). The hematoxylin and eosin (H&E) staining is still considered the gold standard in pathology that has an immense amount of historical data with well-established clinical guidelines (Simonson et al., 2021). With the rich molecular features, linking the MxIF images with the more broadly used H&E images might provide new insights for clinical research. Unfortunately, the H&E images are typically unavailable with MxIF on the same tissue section due to the physical restrictions in imaging procedures (Berens et al., 2019). As the biological H&E staining is not feasible, generative adversarial networks (GANs) have been applied for obtaining virtual H&E (vH&E) whole slide image (WSI) via deep learning empowered virtual staining (Zhang et al., 2020; Nadarajan and Doyle, 2020; Bayramoglu et al., 2017). However, the prior arts did not explicitly model the uniquely rich structural information in MxIF. Some research works also demonstrated MxIF inter-modality set synthesis (i.e., more than 10 stains) (Saurav et al., 2022; Bao et al., 2021b, 2022), but their models were trained with paired MxIF data, and H&E was not involved.

In this work, we propose a novel unpaired "set to image" synthesis model (MxIF to H&E) by utilizing 27 MxIF channels (Figure 1A). Briefly, we extend the CycleGAN (Zhu et al.,

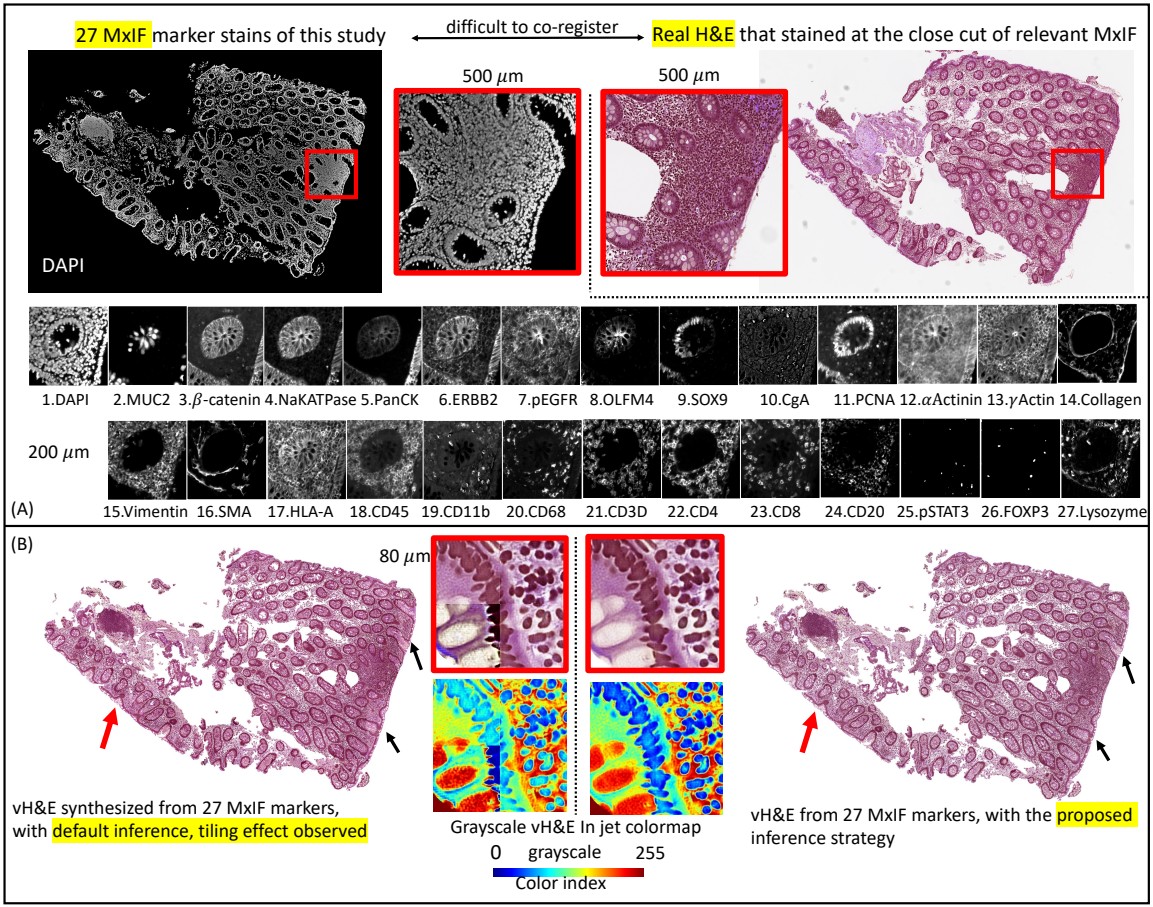

Figure 1: This figure shows the problem setting of the unpaired "set to image" synthesis and the tiling effects in WSI synthesis. (A) We propose to use all 27 markers (set) to synthesize the virtual H&E images (vH&E). (B) The left panel shows that the default patch-wise inference method causes the tiling. The right panel shows that our new method alleviates solve the tiling effect.

2017) framework with a multi-channel synthesis network with extra semantic segmentation branches to ensure anatomical consistency across modalities.

Another long-lasting issue for deep learning based synthesis in histology is that patch-wise training is inevitable for the high-resolution WSIs (de Bel et al., 2018). As shown in Figure 1B, discontinuous intensities are visually identified along the patch boundaries when assembling the individual patches back to a WSI. To address this problem, Lahiani et al. refined the CycleGAN to reconstruct H&E WSIs from FAP-CK images and reduce the tiling artifact by adding the encoder level consistent loss (Lahiani et al., 2020). However, given the domain shifts on H&E RGB images and 27 multi-channel MxIF grayscale images, it is challenging in practise to impose similarity constraints between the encoder of MxIF to

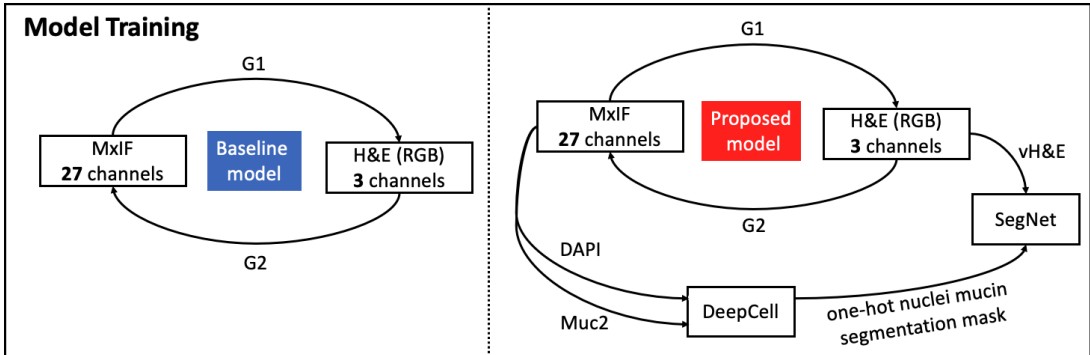

Figure 2: This figure presents the baseline and proposed models for the MxIF and vH&E unpaired synthesis. The baseline models take all 27 channels as a set to synthesize the H&E image. A structural consistent branch (SegNet) is introduced that takes vH&E as input to conduct the nuclei (DAPI) and mucin (Muc2) segmentation.

H&E and the encoder for H&E to MxIF. In this paper, we introduce a random walk sliding window shifting strategy during the optimized inference stage to alleviate the tiling effects that is generalizable.

The contribution of this work is four-fold: **(1)** We propose a novel unpaired "set to image" synthesis model (MxIF to H&E) that utilizes the rich structural information in 27 MxIF channels by aggregating simultaneous nuclei and mucin segmentation supervision as spatial constraints; **(2)** We introduce a sliding window shifting strategy during the optimized inference stage so as to alleviate the tiling effects; **(3)** The proposed random sliding window inference method is a plug-and-play module, which can be generalized and used for other high-resolution WSI image synthesis applications; **(4)** We evaluated the synthesis model in a quantitative manner by employing epithelial cell segmentation as a downstream task.

## 2. Methods

### 2.1. Unpaired set to image synthesis

We used CycleGAN as the backbone of the study. Given two synthesis domains, the generators $G_1$ and $G_2$ synthesize fake images to fool the relevant discriminators $D_1$ and $D_2$ to enhance the reconstruction in both ways. As Figure 2 shows, the $G_1$ of the base model has multiple channels for the MxIF markers (denoted as $X$), $G_2$ takes H&E (denoted as $Y$) to synthesize MxIF, so the input of $G_2$ is a regular three channels for RGB images. The output of $G_1$ is three channels, while the output of $G_2$ is 27 channels. Since the $G_1$ and $G_2$ have incompatible output channels, the total loss we use for the base model is defined in equation 1 without identity loss (Zhu et al., 2017):

$$L_{base} = L_{GAN}(G_1, D_2, X, Y) + L_{GAN}(G_2, D_1, X, Y) + w_{cycle}L_{cycle}(G_1, G_2) \qquad (1)$$

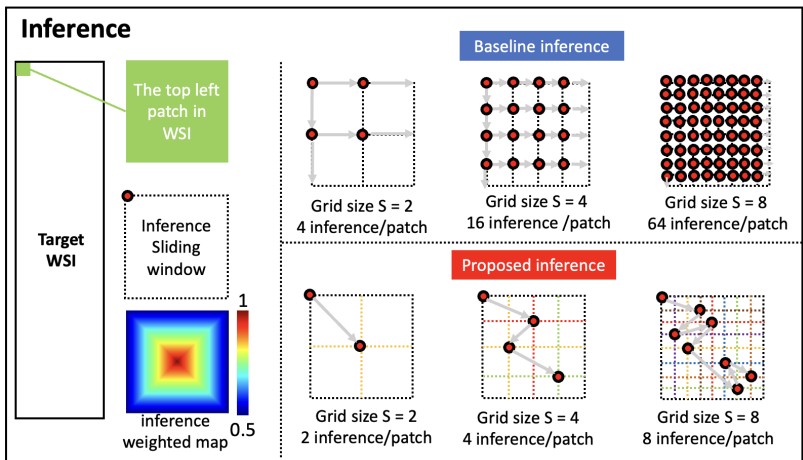

Figure 3: The baseline inference setup and proposed random sliding window inference way. We fix the size of the inference sliding window, and the weighted map is used to average the different inferences. We start the inference from the left top corner of the WSI, then we split the target patch by a grid with size as $S$, the baseline inference will cost $S^2$ inference/patch with a pre-defined step, and the proposed inference will take $S$ inference/patch.

where $L_{GAN}$ is the adversarial loss and $L_{cycle}$ is the cycle consistency loss, $w_{cycle}$ is a constant weight value equal to 10, as referenced in the original CycleGAN article (Zhu et al., 2017). Because the pattern and shape of the nuclei are the critical factors for H&E interpretation, to ensure the structure consistency for each nucleus, and inspired by the Synseg-net (Huo et al., 2018b,a), we added a Res-net based segmentation network (SegNet) that took the vH&E as input (Figure 2). The target reference truth labels were generated from the pre-trained model DeepCell using the DAPI stains and the Muc2 stains (Bannon et al., 2021; Greenwald et al., 2022). We converted the DeepCell instance labels to one-hot semantic labels (denoted as $Z$) to ease the segmentation task. Then, the final loss for the proposed MxIF and H&E refining CycleGAN model is defined in equation 2:

$$L_{propose} = L_{base} + L_{DICE}(G_1(X), Z) \qquad (2)$$

where $L_{DICE}$ is the DICE loss to penalize the wrong prediction (Shen et al., 2018).

## 2.2. Random sliding window for inference

(de Bel et al., 2018) proposed using overlap patches to generate inference of the adjacent tile to prevent artifacts at the edges of tiles. To create the final inference result of the overlap sections, (de Bel et al., 2018) claimed to calculate the sum of the weighted pixel values and divide by the sum of the weights. Inspired by (de Bel et al., 2018), we further define the pixel close to the center of the sliding window would contribute 100%, and the pixel in the inference sliding window boundary would contribute 50%, as shown in Figure 3. To avoid

confusion, we only make the sliding window move in the direction of right or down. The region of the sliding window outside the target patch would be weighted average by relevant adjacent patches. We define the inference sliding window's size as $W_s$. The issue of (de Bel et al., 2018) is computationally expensive. For instance, if the pre-defined moving step of the sliding window is $step = \frac{1}{2}W_s$, then the target patch is split as a grid with a size $S = 2$. The base model would traverse all possible $S^2 = 4$ nodes. Similarly, if $step = \frac{1}{4}W_s$, then the grid size is $S = 4$, and the base model would use $S^4 = 16$ inferences per patch.

In our proposed approach, for the first patch of the whole WSI (the top left one), if the grid size $S$ is pre-defined, then we downgrade the seeking of the potential inference point as a random walk problem, which means the inference sliding window start moving from the top left corner and it should not visit any inner grid lines twice. Then, we randomly select a potential $S$ node as a set as

$$R = \{(x_0, y_0), ..., (x_{s-1}, y_{s-1})\} \tag{3}$$

where $(x, y)$ is a node in the grid, $(x_0, y_0)$ is always the top left corner of the WSI. Obviously, the size of $R$ is $S$. For any two nodes $(x_i, y_i)$ and $(x_j, y_j)$ in $R$, $x_i \neq x_j, y_i \neq y_j$. Including the randomness on sliding window has been proven effective for model training, especially on 3D medical images (Tang et al., 2021, 2022). Very few, if any, studies have been focused on utilizing the random shifting window for medical image inference. Once the $R$ is finalized, we keep moving sliding window by $step = W_s$ horizontally and vertically from any node in $R$. The main motivation behind the random design is to ensure that multiple sliding windows would cover each pixel of the tissue in the WSI under a statistical fashion. The proposed way reduces the inference computation complexity from $O(S^2)$ to $O(S)$ per patch.

## 3. DATA AND EXPERIMENTAL DESIGN

### 3.1. Data and preprocessing

The data were collected from in-house patients with Crohn's disease (CD), a challenging inflammatory bowel disease (IBD), which is characterized by chronic relapsing and remitting bowel inflammation and causes tremendous intestinal inury(Baumgart and Sandborn, 2012). 28 biopsies (14 ascending colon (AC) tissue samples, 13 terminal ileum (TI) samples) were collected from 16 CD patients, and 4 healthy controls with IRB approval (Bao et al., 2021a). The MxIF markers were stained and acquired from three batches with $20\times$ magnification. The markers are DAPI, Muc2, $\gamma$Actin, CD45, CD11B, Collagen, CD20, PCNA, $\beta$-catenin, pSTAT3, pEGFR, CgA, CD4, CD3d, HLA-A, PanCK, OLFM4, CD8, $\alpha$Actinin, CD68, NaKATPase, Vimentin, Sox9, FOXP3, Lysozyme, SMA, and ERBB2. Each MxIF marker images was preprocessed by DAPI-based spatial alignment, autofluorescence correction, and background masked-out (McKinley et al., 2017). 28 H&E WSI images were used from the adjacent section close cut of MxIF tissue sections (more detail of dataset description is listed in Appendix-B).

### 3.2. Experimental Design

We set up the training testing $8 : 2$ split-ratio and considered sample diversity, where 22 samples were used for training while 6 were tested. We crop all WSIs into $1024\times1024$ patches

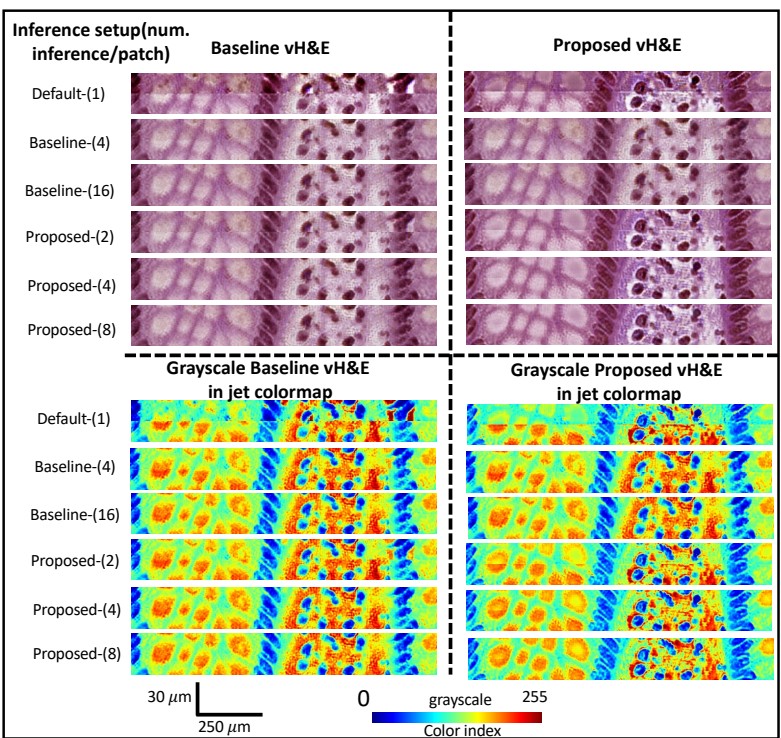

Figure 4: The qualitative results of titling effect removal on a random selective ROIs. Both vH&E in raw intensity and grayscale vH&E in jet colormap are shown.

(0.324 $\mu$m/pixel) while only keeping the ones with 80% and more foreground tissues. Finally, 265 patches were remained for MxIF and 181 for H&E. The experiments were performed on a workstation with an NVIDIA RTX A6000 GPU.

**Tiling effect evaluation.** We infer the testing samples using six ways. The first is default inference uses one inference. The second to fourth approaches use the proposed method with different grid sizes ($S = 2, 4, 8$). Then the proposed method would use 2, 4, and 8 inferences per patch. The fifth and sixth inference methods use the baseline when $S = 2$ and 4, which causes 4 and 16 inferences per patch. For computation efficiency, we do not perform baseline when $S = 8$. To validate the effect of random walk, we validate the proposed method when $S = 4$, where $3! = 6$ cases are available.

To evaluate the tilting effect removal performance, we find all lines of longitude and latitude if the coordinates of lines can be divisible by 128. Then we collect the pixels of two sides of the lines with one width or height, convert the RGB values to CIELAB space, and calculate the Euclidean distance of relevant two adjacent stripe pixels along the lines. In summary, there are 1521 lines across all six testing samples are collected for validation.

**vH&E for downstream task.** To further quantitatively evaluate the overall synthesis performance, we performed downstream epithelial cell segmentation experiments using

Table 1: Results for the tiling effect removal mean (standard deviation) Euclidian distance values from different inference methods. The best strategy is highlighted with a significant difference from other ways by the Wilcoxon signed-rank test ($p < 0.05$).

| | | vH&E base model | vH&E proposed model |
|---|---|---|---|
| **Validation of** | Default-(1) | 460.6 (408.4) | 489.2 (398.2) |
| **different** | Baseline-(4) | 383.2 (237.1) | 413.4 (243.9) |
| **setup** | Baseline-(16) | 356.2 (200.2) | 386.8 (213.1) |
| **(num. of** | Proposed-(2) | 397.9 (252.9) | 425.7 (254.5) |
| **inference** | Proposed-(4) | 366.0 (206.0) | 394.7 (217.7) |
| **per patch)** | **Proposed-(8)** | **349.7 (193.4)** | **379.2 (207.6)** |
| | | **vH&E base model** | **vH&E proposed model** |
| | Proposed-(4) case1 | 366.0 (206.0) | 394.7 (217.7) |
| | Proposed-(4) case2 | 365.7 (206.2) | 394.9 (217.8) |
| **Effect of** | Proposed-(4) case3 | 365.6 (205.8) | 394.6 (217.5) |
| **random walk** | Proposed-(4) case4 | 366.1 (206.1) | 395.3 (217.8) |
| | Proposed-(4) case5 | 366.6 (206.3) | 394.8 (217.8) |
| | Proposed-(4) case6 | 366.6 (206.4) | 395.3 (217.9) |

vH&E. To generate the reference label, we manually select the threshold to find definite positive signals of (NaKATPase+ PanCK+) and exclude Vimentin+. Then we integrate such MxIF biological rule-based threshold knowledge to annotate the DeepCell instance segmentation label and convert the annotation to semantic labels as the training and testing annotations. We crop the WSI into 256×256 patches (0.5 $\mu$m/pixel). We run the pretrained HoVerNet model for the CoNIC challenge (Graham et al., 2019, 2021) on the 319 vH&E patches and get the generated epithelial labels. We utilize the masked DICE coefficient score for epithelial cell segmentation label consistency from MxIF and vH&E.

## 4. RESULTS

**Tiling effect evaluation.** Figure 4 shows the qualitative results of titling effect removal on a random selective region of interest (ROI). Both vH&E in raw intensity and grayscale vH&E in jet colormap are shown. The quantitative results are listed in Table 1, where we found that for each of the two vH&E syntheses, the proposed method with grid size $S = 8$ outperformed the best baseline setup with 50% less computation usage. Table 4 also presents the effect of random walk, where no significant difference is found.

**vH&E for downstream task.** Figure 5 shows qualitative and quantitative results for the two vH&E syntheses on epithelial cell segmentation consistency between DeepCell and MxIF biological rule-based label and H&E HoverNet label. Two sample ROIs from testing AC and TI samples are randomly selected. The results show that the proposed CycleGAN with structure constraints outperforms the baseline by 56% improvement.

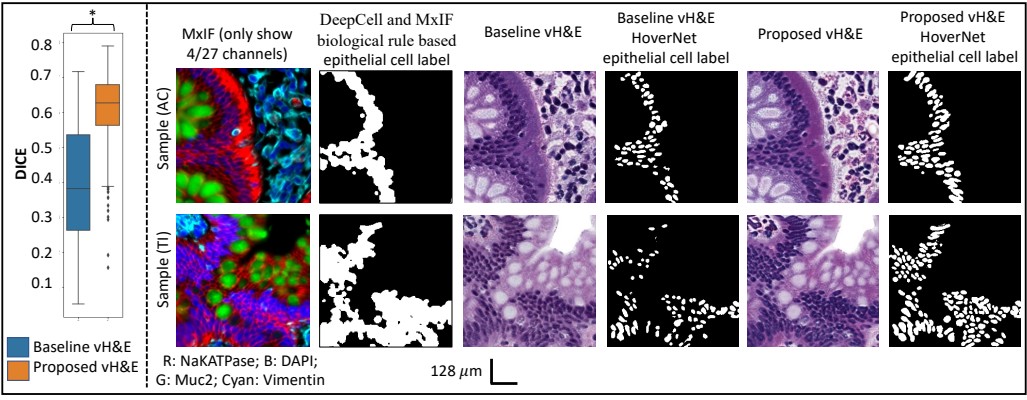

Figure 5: The performance of the downstream epithelial cell segmentation task using vH&E synthesis. The left panel shows the epithelial cell segmentation performance with significance found by the Wilcoxon signed-rank test ($p < 0.05$, marked as *). The right panel shows the qualitative results between the DeepCell and MxIF rule-based label versus the vH&E HoverNet label.

## 5. Discussion and Conclusion

Direct evaluation on image similarity is essential. However, we do not have paired MxIF and H&E data for such direct comparison. A downstream epithelial cell segmentation task would be helpful for evaluating the quality of synthesis only, which is a main limitation of our work. DeepCell nuclei segmentation is prone to over-segmenting the nuclei, which could introduce bias in the downstream analysis when we compare the results between the MxIF DeepCell label and the HoverNet label. A refined MxIF label could potentially improve validation accuracy that requires further investigation.

In this paper, we present a novel unpaired "set to image" synthesis model (MxIF to vH&E) that fully utilizes the rich structural information in 27 MxIF channels by adding simultaneous nuclei and mucin segmentation supervision as spatial constraints. A random sliding window shifting strategy is introduced during the optimized inference stage torwards alleviate the tiling effects. Our simple inference method is a plug-and-play module that can be generalized and used for other high-resolution WSI image synthesis applications. Our model our proposed vH&E synthesis method achieves superior performance compared with the baseline approach.

## Acknowledgments

This research was supported by The Leona M. and Harry B. Helmsley Charitable Trust grant G-1903-03793 and G-2103-05128, NSF CAREER 1452485, NSF 2040462, and in part using the resources of the Advanced Computing Center for Research and Education (AC-CRE) at Vanderbilt University, Nashville, TN. This project was supported in part by the National Center for Research Resources, Grant UL1 RR024975-01, and is now at the Na-

tional Center for Advancing Translational Sciences, Grant 2 UL1 TR000445-06, the National Institute of Diabetes and Digestive and Kidney Diseases, the Department of Veterans Affairs I01BX004366, and I01CX002171. The de-identified imaging dataset(s) used for the analysis described were obtained from ImageVU, a research resource supported by the VICTR CTSA award (ULTR000445 from NCATS/NIH), Vanderbilt University Medical Center institutional funding and Patient-Centered Outcomes Research Institute (PCORI; contract CDRN-1306-04869). This work is supported by NIH grant T32GM007347 and grant R01DK103831. We extend gratitude to NVIDIA for their support by means of the NVIDIA hardware grant.

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

## Appendix A. Supplementary information on tiling Effect

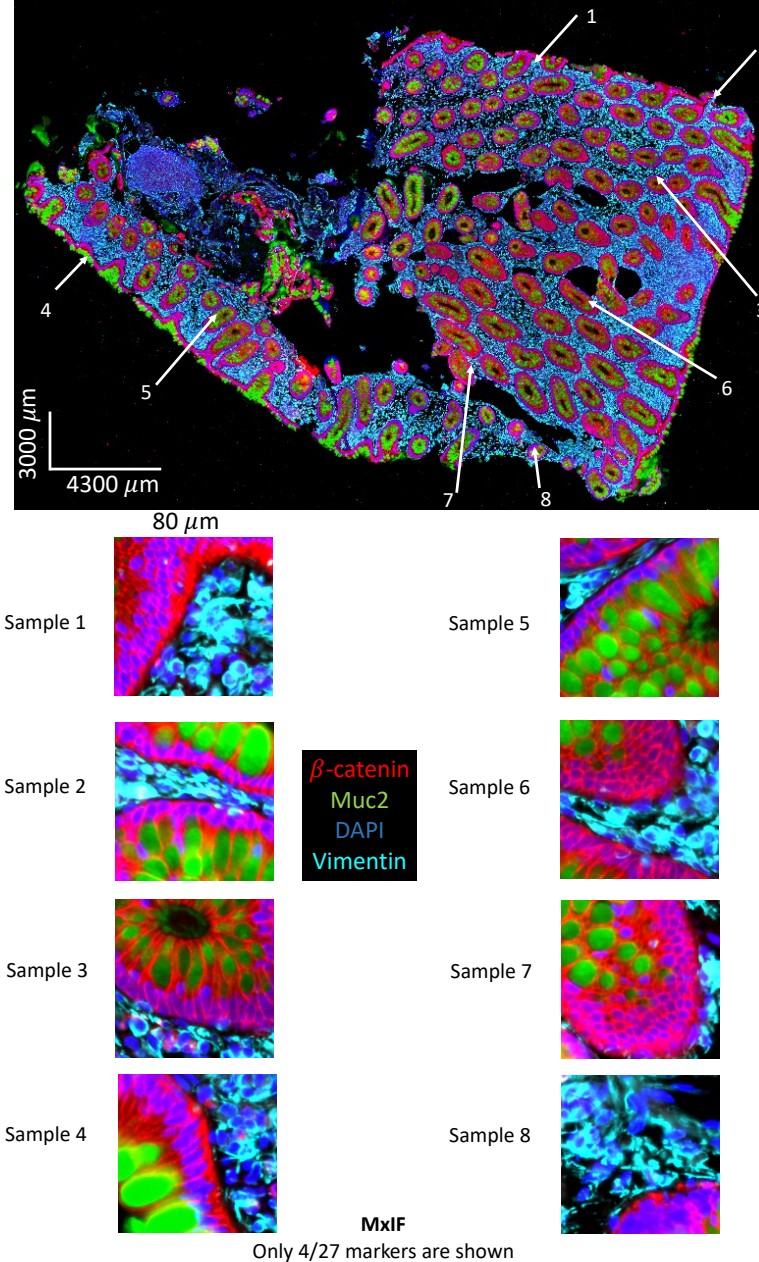

Figure 6: Selective field of view on MxIF WSI data. Only 4 (out of 27 stains) markers are chosen for visualization. The intensity are adjusted for illustration purpose.

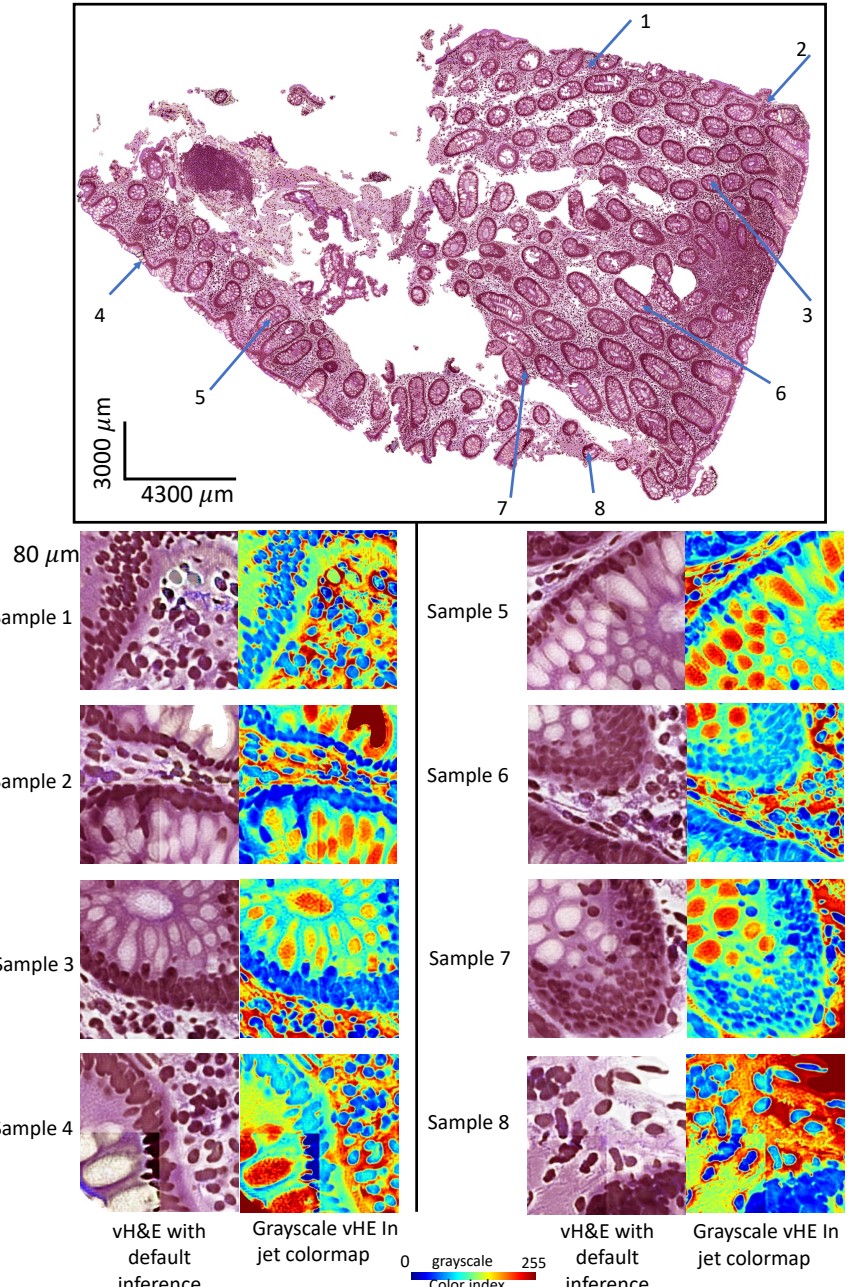

Figure 7: Selective bad examples of the tiling effect in vH&E after stitching the patch back into WSI using default inference method with above MxIF data as input.

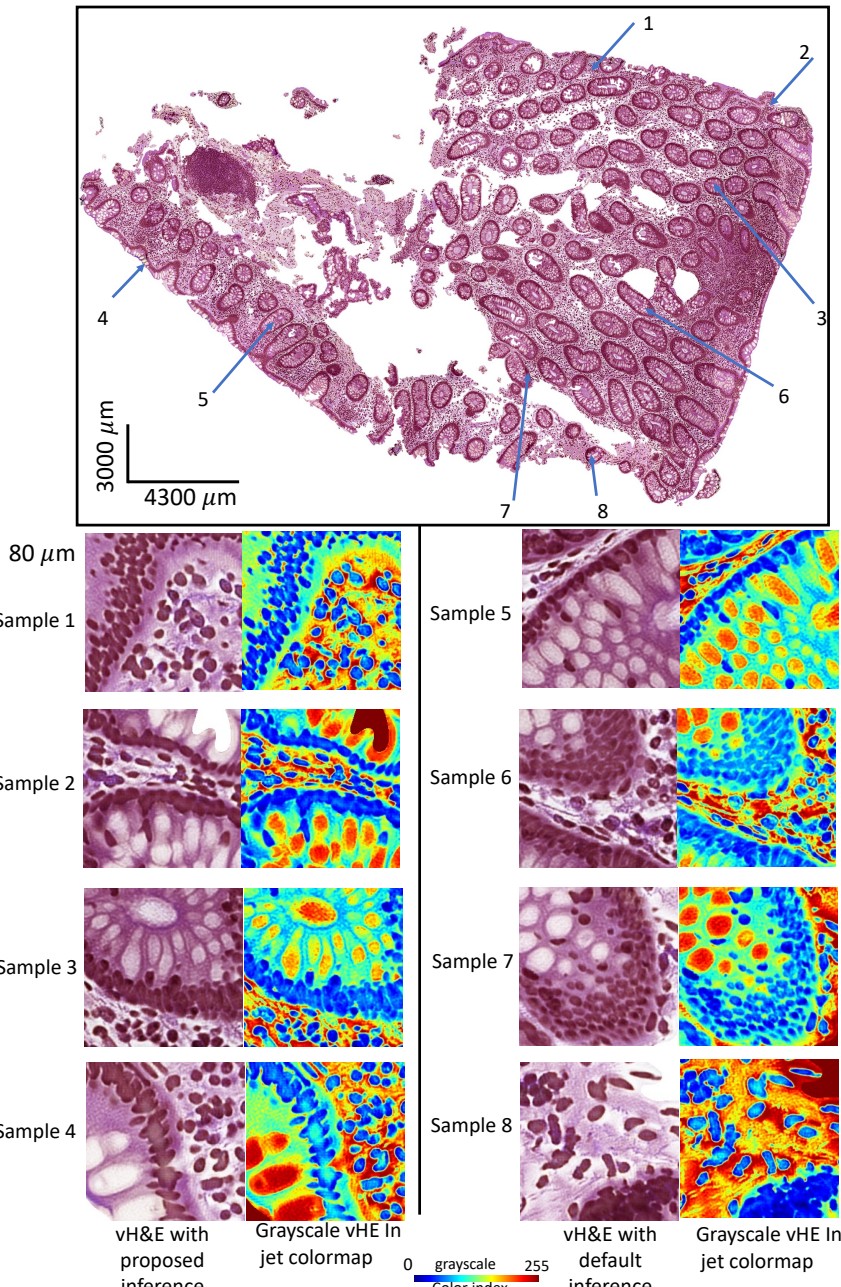

Figure 8: Same field of view with the tiling effect removal in vH&E using proposed inference method.

## Appendix B. Supplementary detail of random walk inference

**How to guarantee random walk can cover all pixels of WSI.** For default inference, the WSI is divided using a sliding window with a static step, where the step size is equal to the size of the sliding window. This ensures that all divided patches have no overlap, and every pixel of the WSI is covered. The proposed method uses the default inference as the base inference for further averaging. Additionally, the proposed method only shifts the starting position of the first top-left corner of the sliding window in the WSI, as shown in Figure 3. Starting from the top-left corner, we randomly walk the graph and select a few nodes as potential starting positions for the sliding window, ensuring that none of the nodes share the same x or y coordinate. Once the nodes are finalized, for each node, we allow the sliding window to move with a static step equal to the size of the sliding window, using the default inference method.

For instance, to get the final intensity values per pixel, if the grid size is 8, then there will be an 8 potential start position of the first sliding window in one random walk. After each sliding window traverses the WSI, every pixel will have 8 values to be averaged, except for the leftmost column and the topmost row of the patches WSI, where each pixel may have 1 to 8 potential weights for averaging. However, these areas are usually empty glass. If there is indeed tissue in those areas, zero padding can be applied to the WSI to ensure that all pixels that contain tissues have 8 values for the weighted average.

**How inference map is created and used.** The weight for a single pixel is based on the following formula is based on

$$w = -max(|x - x_{cp}|, |y - y_{cp}|) \rightarrow linear\ normalize\ to\ [0.5, 1] \tag{4}$$

where (x,y)is the coordinate of the pixel within a sliding window, $(x_{cp},\ y_{cp})$is the center coordinate of the sliding window. The final value of a pixel in a target coordinate is calculated by the sum of all inference values of the pixels divided by the sum of weights.

## Appendix C. Supplementary detail of dataset description

The dataset was collected from an ongoing clinical study to understand Crohn's disease. The tissue biopsies of patients were obtained from the ascending colon and terminal ileum, with a focus on epithelial cell nuclei as a proof of concept. Epithelial cells are one of the important cell types in the colon area for Crohn's disease. The slides used for MxIF were cut at a different time than those used for the H&E staining. Typically, a formalin-fixed paraffin-embedded tissue section placed on a slide is a 5um thick cut. When cutting new sections from a tissue block, the first section is usually not usable, and it may take a few cuts to get intact sections to put on a slide. The exact distance between the MxIF and H&E slides was not recorded in the clinical data acquisition. To define the dataset more precisely, we ensured that the section of MxIF and the section of H&E were adjacent sections. However, based on our empirical image registration, either rigid or non-rigid, on the closely cut MxIF and H&E tissues, the results at the cellular level were poor. Figure 1 shows that the cell pattern between MxIF (top left) and the real H&E (top right) did not match at the cellular level (although the shape of the WSI seems to match), making it almost impossible to align the MxIF and H&E in our dataset.

The MxIF data was acquired using a 20× magnifying camera that generated images with a physical size of 0.324 $\mu$m per pixel. For training the CycleGAN, 265 MxIF patches with a patch size of 1024×1024 pixels and 0.324 micron per pixel were used. To perform downstream analysis, we needed to leverage the results from the HoverNet, which was trained on H&E 20× images with a physical size of 0.5 $\mu$m per pixel. Therefore, we had to downsample the virtual H&E (vH&E) images to the HoverNet space. Additionally, the input patch size for HoverNet is 256×256 pixels, and we only kept patches that contained over 10% epithelial cells as determined by the MxIF biological rule-based labels. This is how we ended up with 319 vH&E patches for testing in the downstream analysis.

## Appendix D. Supplementary detail of downstream analysis

There are two types of MxIF labels used in this work: (1) we use the DeepCell pre-trained model to segment DAPI/Muc2, and convert the initial instance segmentation (denoted as $L_S$) output to a semantic mask ($L_{SN}$) for CycleGAN training only. (2) For downstream analysis, we annotate the $L_S$ and convert it to a semantic mask as epithelial cell labels (denoted as $L_{SE}$), which will then be further compared with the output labels from the HoverNet models. To prove the usefulness of the vH&E, the HoverNet pre-trained model could segment the vH&E, generate nuclei instance segmentation mask, and the HoverNet can annotate the type of the nuclei submontanely. We converted those instance nuclei labels and created a semantic mask as $L_H$. Finally, we can utilize the HoverNet semantic epithelial mask $L_H$ to compare with the $L_{SE}$ for evaluation.

**Limitation of downstream analysis.** Direct evaluation of image similarity is essential. However, we do not have paired MxIF and H&E data for such direct comparison. A downstream epithelial cell segmentation task would be helpful for evaluating the quality of synthesis only, which is a main limitation of our work. DeepCell nuclei segmentation is prone to over-segmenting the nuclei, which could introduce bias in the downstream analysis when we compare the results between the MxIF DeepCell label and the HoverNet label. A refined MxIF label could potentially improve validation accuracy that requires further investigation. It is also essential to perform the tiling approach on the downstream task. Given that the downstream analysis only focuses on epithelial nuclei segmentation and the limited epithelial cell sitting on the tile edges, we should include this experiment design for the further extension of this work with a larger scale dataset.

