# OpenReview forum: "Alleviating tiling effect by random walk sliding window in high-resolution histological whole slide image synthesis"
_MIDL.io/2023/Conference — MIDL 2023 Poster_

### Official Review · Reviewer_WSbG · 2023-02-02

**Confidence:** 3
**Preliminary Rating:** 4
**Recommendation:** Poster

**Summary:**

The authors describe a novel scheme to translate multiplex immuno-fluorescence imaging into H&E images by the means of using unpaired image-to-image translation (CycleGAN architecture). The main novel twist in their work in the combination with an additional, IHC-derived, segmentation loss to combat inconsistency across tiles when generating a new WSI out of the MxIF images. Further, they propose and evaluate a novel less computationally demanding strategy for tiling/sliding over the original image to create the synthesized WSI.

**Strengths:**

- I think the method is really clever and the paper is generally well written.

- In my view it represents a very interesting extension of CycleGAN and addresses the typical weaknesses (factual inconsistency, e.g. by inverting information) of the approach well.

- I also found the approach of random patch route selection interesting.

- The evaluation is performed against a suitable baseline


**Weaknesses:**

- Since you trained with cell segmentation as a target task, it could well be debated if the same task is a suitable proxy metric for the evaluation. Ultimately, your target is creation of virtual H&E and the quality of this is only indirectly linked to the ability to perform a rather simple task (cell segmentation). As such, I would see a (maybe even subjective, expert) evaluation a valuable add-on to see if the proxy metric actually is strongly linked with image quality. Maybe even a diagnostic downstream task that is unrelated to cell segmentation would be viable in this sense.

- The paper is a bit dense with information for a conference paper, which reduces readability a bit. Don't get me wrong: I do like the paper and the method that is presented, but I think a little more glue within the sections would have helped to understand it better. In particular, I think section 2.2 could use a little more introduction. As a scholar not familiar with the works of de Bel et al., I had a hard time getting into what the authors did there. Instead, the paper is completely missing a discussion of the results, which is a pity.

- It would be really good if the authors could provide access to their code and data. As far as I know there is little MxIF datasets out there and this can only be changed if authors volunteer to publish their data.

**Deanonymize Review:**

no

**Detailed Comments:**

- I am not an expert in MxIF so I don’t know i this is feasible, but we used multiple IHC stains (Ki-67, PHH3) with tissue, where we initially stained with H&E (and scanned with an WSI scanner) and then re-stained with the IHC stain with quite some success. Could this be viable for quality assessment for your work as well?

- At one instance the authors used the word "tilting“ instead of "tiling"

- Please really think also about discussing also weaknesses of the approach. At least in my own experience each approach has its imperfections and as the authors you are highly knowledgeable in this domain. Other researchers will find it very insightful if these are also discussed.

**Paper Type:**

methodological development

**Questions To Address In The Rebuttal:**

- Please provide a rationale of why the downstream task (cell segmentation) is really suitable for the evaluation of your method (which, as stated, aims at image translation, not at creating segmentation models). This is especially important since it was already used within the network training.

- Please provide a rationale of why it was not possible to publish code and data alongside your work.

---

### Official Review · Reviewer_hBzx · 2023-02-03

**Confidence:** 4
**Preliminary Rating:** 3
**Recommendation:** Poster

**Summary:**

The paper at hand deals with the translation of multiplex immunofluorescence images to H&E-stained images using a CycleGAN to leverage existing knowledge and guidelines for H&E image data. The proposed method includes a segmentation branch to improve content consistency. Additionally, a sliding window method with a topological sorting approach is used to compensate for tiling effects caused by the patch-wise processing of large images (WSIs). The authors evaluate the improvement compared to a standard CycleGAN on the down-stream task of cell classification and report considerably improved results.

**Strengths:**

- Smart and simple idea to reduce computation time and tiling effects for large images which can easily added post-hoc and is modular
- Quantitative evaluation on the down-stream task of nuclei segmentation
- The target application is a suitable and well-motivated use case of CycleGANs for unpaired image translation

**Weaknesses:**

- The evaluation of resulting image quality is not really targeted for the goal of the task of H&E synthesis (just on a downstream task, on a potentially mediocre ground truth, which is not discussed). The impact of the tiling approach on the downstream task is not assessed.
- Motivation and description not always really clear (e.g., cell classification vs. cell segmentation, dataset description, see also detailed comments)
- The main argument for not using the method by Lahiani et al. (which has even faster runtimes) is that consistency is difficult to achieve for 27 channels; however, no quality assessment or other evaluation is performed for this step, and since the generation of H&E images is in the focus of this work, the consistency loss could simply be used only for the H&E generation.
- The input data seems to be very high-dimensional for the task, especially given the low number of samples. The need to use 27 markers for the translation to H&E (and why it makes sense to transform the H&E image to a 27 channel-MxIF) is not well motivated and not further validated.

**Deanonymize Review:**

no

**Detailed Comments:**

Short comment before the rest of the review: A thank you to the authors for not having an extensive supplementary material section but instead submitting a self-contained paper.

In the following, I would like to address some points mentioned above, provide additional detail, and raise additional points.

The proposed "random sliding window strategy" can be a simple but elegant way to reduce the number of inferences while keeping image quality. The experiments show reduced tiling artifacts with the same or lower computational effort. The idea behind this approach (and the difference to de Bel et al.) could be highlighted and advertised more strongly in the abstract and in the introduction. Here, the term "topological sorting" could be even more appropriate than "random" since the placement is not fully random across the grid. Still, there are also a couple of aspects that I am not fully clear about here:
- I don't fully understand the experimental setup: How large were the images that were "tiled back together"? I am unclear on that, since the authors mention cropping the WSI into tiles of 1024x1024. For input images sized 1024x1024, a since inference step (without any tiling) should be feasible on an A6000 graphics card, therefore I don't expect any tiling here. Still, Default (1) has the highest error. How exactly is the problem setup here?
- I don't really understand the goal of the ablation study - what exactly is ablated?
- The authors mention the work by Lahiani et al., which has an even lower runtime (O(1)), and claim that it is difficult to ensure consistency for 27 channels. Since it is not the goal of this paper to generate MxIF images, the tiling effect would not be relevant for this translation and the additional consistency loss would only have to be applied for the MxIF encoder. Independent of the beauty and benefit of the post-hoc use of the random-sliding-window approach, I am missing further justification why Lahiani et al. is not used as a reference method in the quantitative comparison.

Fundamental question: I do not fully understand the motivation behind using 27 highly specific markers to predict an H&E image, especially given the relatively small, in-house dataset (20 patients/28 samples). Why didn't the authors use a sensible subset with suitable morphology-related IF markers for their image-to-image translation? I see the major problem here not in MxIF-->H&E step (though there is likely a lot of superfluous information potentially resulting in overfitting). I believe that this can be problematic in the backward step, since I find it rather implausible that an H&E image can be mapped to a 27 MxIF image. Especially for the "cycling" MxIF-->H&E-->MxIF, there is a pretty high chance that the model is going to store the corresponding information undetectable for the discriminator (GAN steganography, e.g., arXiv:1712.02950, 2017) to be able to recover the MxIF image. The high-frequency information in Fig. 4 could point toward this.

Regarding the auxiliary task and the validation on the downstream task: I understand the issue with not having paired images for evaluation; however, I see multiple problems with the approach the authors presented.
- Minor: the authors sometimes use cell classification and sometimes cell segmentation - this should be ironed out
- The authors mention that close sections (5$\mu$m apart) of H&E and MxIF images are available - I would therefore expect that at least a general qualitative, regional assessment of image quality and larger structures is feasible
- The evaluation on the down-stream task is interesting and I understand that it is a crutch in the absence of missing data, but
    - The down-stream task makes fairly little sense to me since the problem is defined on the MxIF image - why would it be interesting to do a nucleus segmentation on an vH&E image if the DAPI/Muc2 stain is available? This is not discussed.
    - The authors use the same task for validating the improved performance of their network and for regularizing their network - it would be very counter intuitive that the so-trained network performs worse on this task.
    - Looking at Fig. 5: The nuclei seem to heavily over-segmented in epithelial cell labels and therefore provides fairly weak evidence that there is a sensible cell-wise translation from MxIF-->H&E. This is not discussed or put into context.

A complementary validation scheme that does not use the same task twice and/or a task that is not directly related to one of IF markers used for synthesis could have been a more reliable and relevant down-stream task.

Additional suggestions for future work:
- specific IF markers are known to be "brittle" or to have a higher variance - investigating the robustness to changes could be interesting

**Minor:**
- Short title (header) is missing

p2:
- why are there yellow text markings in Fig. 1?
- "with multi-channel synthetic network" - "with a ..." (typo), also what is meant with "synthetic network" here?

p3:
- "However, ensuring 27 channels of MxIF latent space is as consistent as 3 channels H&E is challenging in practise." - I am not sure I understand what this means; this sentence and the following would benefit from additional text revision (I would avoid extensive use of "so as to")

p4:
- "that equals 10 references from the original CycleGAN article" - rephrase?
- Fig. 3: How do the authors arrive at this weighting map? Does this make sense for a tiling based on random shifts? Why wouldn't you simply want to average over all signal provided for a specific pixel?

p6:
- Fig. 4: The rule at the bottom does not make sense with 10 vs. 200 $\mu$m; the figure is also not fully clear to me: Does "proposed-(x)" refer to the tiling scheme and "proposed vH&E" refer to the additional Dice loss?
- was a patient-wise train/test split observed?
- can the authors specify what they mean by "we ... considered sample diversity"?
- I would appreciate a short additional explanation as to why the evaluation is done in LAP
- A600 --> A6000 ?

p7:
- why 319 vH&E patches? Previously, 265 patches for MxIF were mentioned (does this refer to the train or the test set?)

p8:
- Fig 5: I would again doubt the ruler, given the size of the nuclei


**Paper Type:**

methodological development

**Questions To Address In The Rebuttal:**

I would like the authors to address mainly the following points in their rebuttal:
- A more detailed explanation of the experimental setup with respect to the tiling (see above)
- A justification for using all 27 channels
- A discussion with respect to the quality and a potential bias in the ground truth masks and a discussion why *no* qualitative assessment was conducted although at least close sections (H&E and MxIF) were available

---

### Official Review · Reviewer_w8oD · 2023-02-06

**Confidence:** 3
**Preliminary Rating:** 3

**Summary:**

This paper aims to transform MxIF WSIs into virtual H\&E stained WSIs. The baseline model is basically a CycleGAN, and the authors proposed to add a segmentation branch to utilize the rich structural information, in order to further enhance the quality of the synthetic images. The evaluation shows the proposed method can alleviate the tiling effects and reduce the computation complexity.

**Strengths:**

* Well-written.
* As a whole slide image has a huge resolution, the synthetic process has to be based on patch level, so the tiling boundary effect should be carefully considered. Thus, this paper is working on an important problem.
* The segmentation branch is an interesting idea


**Weaknesses:**

* The random sliding window part is kind of hard to follow. Maybe providing more details and figure illustration can address this problem.
* The segmentation part should also be included in the title if it is also the contribution of this paper.
* The motivation behind such a random design is not very clearly. Perhaps authors can elaborate more on the description.


**Deanonymize Review:**

no

**Paper Type:**

methodological development

**Questions To Address In The Rebuttal:**

* How can it be guaranteed the random sliding window can traverse every corner in a WSI?
* How does the inference weighted map be used?  Can the math expression be provided, please?
* Authors are suggested to provide some bad examples of the tiling effect after stitching into WSI.

---

### Meta-Review · Area_Chair_3RSJ · 2023-02-24

**Recommendation:** Accept (Poster)
**Confidence:** 5

**Metareview:**

The paper proposes a method to transform multiplex immunofluorescence images into H&E-stained images using a CycleGAN and a segmentation branch to enhance content consistency. A sliding window method is also used to address tiling effects caused by patch-wise processing of large images. The proposed method shows considerable improvement in downstream cell classification compared to a standard CycleGAN.

This paper presents a well-written and clever approach to using CycleGANs for unpaired image translation of whole slide images. The addition of a segmentation branch to reduce computation time and tiling effects for large images is a smart and simple idea. The paper addresses the weaknesses of CycleGAN well and the evaluation is performed against a suitable baseline, with a quantitative evaluation on the downstream task of nuclei segmentation.

The reviewers raised a number of weaknesses related to this work. The paper is criticised for being dense with information and lacking a clear motivation behind the use of a random sliding window approach. The segmentation part should be included in the title if it is a contribution, and the impact of the tiling approach on downstream tasks is not assessed. The authors are also questioned about the need for 27 markers for the translation to H&E and the suitability of cell segmentation as a proxy metric for evaluating the quality of virtual H&E images. Additionally, the paper lacks a discussion of results and would benefit from more introduction in section 2.2. The lack of code and data availability is also noted as a limitation.

The authors have responded to the different remarks of the reviewers, even a second round of remarks for one of the reviewers. The manuscript has been revised accordingly, appears clearer and more relevant now and above all allows to better situate the authors' contributions. I therefore recommend accepting this work for MIDL.